# Temporal and Spatial Activity Patterns of Sympatric Wild Ungulates in Qinling Mountains, China

**DOI:** 10.3390/ani12131666

**Published:** 2022-06-28

**Authors:** Jia Li, Yadong Xue, Mingfu Liao, Wei Dong, Bo Wu, Diqiang Li

**Affiliations:** 1Institute of Ecological Conservation and Restoration, Chinese Academy of Forestry, Beijing 100091, China; lijia2530@126.com (J.L.); wubo@caf.ac.cn (B.W.); 2Ecology and Nature Conservation Institute, Chinese Academy of Forestry, Beijing 100091, China; xueyadong334@163.com; 3Key Laboratory of Biodiversity Conservation of National Forestry and Grassland Administration, Beijing 100091, China; 4School of Foreign Languages, Jiangxi Agricultural University, Nanchang 330045, China; jxaulmf@163.com; 5Changqing National Nature Reserve, Hanzhong 723000, China; pandadongwei@163.com

**Keywords:** coexistence, temporal overlap, camera trapping, relative activity index, efficient management

## Abstract

**Simple Summary:**

Information on species’ niche differentiation will contribute to a greater understanding of the mechanisms of coexistence benefitting the conservation and management of ecological communities. The widespread reduction in apex predators and more restricted hunting management has con-tributed to an increase in the abundance of wild ungulates in the Qinling Mountains, presumably resulting in an intensifying interspecific competition pressure. However, the activity patterns of the species in this region are completely unknown due to difficulty in accessing the locations where they occur. Thus, we used camera trapping to systematically investigate spatial and temporal activity patterns of sympatric ungulates in the Qinling Mountains, where top predators are virtually absent. This intensive camera-trap survey elucidated much more detailed studies of spatial and temporal activity patterns in multiple sympatric wild ungulates under natural conditions. Further, our results provided detailed information of the spatial and temporal ecology of ungulate communities in forest ecosystems, which would be a guide to establishing conservation priorities as well as efficient management programs.

**Abstract:**

Dramatic increases in populations of wild ungulates have brought a new ecological issue in the Qinling mountains. Information on species’ niche differentiation will contribute to a greater understanding of the mechanisms of coexistence, so as to ultimately benefit the conservation and management of ecological communities. In this study, camera trapping was used to investigate spatial and temporal activity patterns of sympatric wild ungulates in the Qinling Mountains of China, where top predators were virtually absent. We obtained 15,584 independent detections of seven wild ungulate species during 93,606 camera-trap days from April 2014 to October 2017. Results showed that (i) the capture rate differed significantly across species, with the capture rate of reeve muntjac being significantly higher than that of other species; (ii) the wild boar had a higher occupancy rates (*ψ* = 0.888) than other six ungulates, and distance to settlements had a negative relationship with wild boar (*β* = −0.24 ± 0.17); (iii) the forest musk deer and mainland serow had low spatial overlaps with other five wild ungulates, while spatial overlap indices of any two given pairs of wild ungulates were relatively high; (iv) all wild ungulates species (expect wild boar) were mainly active during crepuscular and diurnal periods, and showed bimodal activity peaks at around 05:00–07:00 and 17:00–19:00; and finally, (v) all wild ungulates showed moderate to high temporal overlaps. The results provided detailed information of the spatial and temporal ecology of wild ungulate communities in forest ecosystems of China, which also would be a guide to establish conservation priorities as well as efficient management programs.

## 1. Introduction

Understanding the coexistence mechanism between ecologically similar species is an important issue in ecology throughout the world [1,2]. Interspecific competition plays an important role in shaping communities by affecting the ability of component species to access limited resources [3,4]. With reduction in agonistic interactions within communities, species often segregate along one or more dimensions (e.g., spatial, temporal and resource) of their ecological niches in order to promote coexistence, a process known as niche differentiation [5,6]. Information on species’ niche differentiation can contribute to a greater understanding of mechanisms of coexistence so as to ultimately benefit the conservation and management of ecological communities [7,8].

Wild ungulates affect ecosystem structure and function, and can serve as an important ecological indicator of the terrestrial ecosystem health [9]. Wild ungulates are highly integrated with components of grassland food webs that exert strong direct and indirect influences on vegetation composition, which may alter plant communities through the extensive grazing, browsing, trampling and defecation [10], not only shaping the structure and distribution of the vegetation, but also affecting nutrient flows and the responses of associated fauna [11,12,13]. Additionally, wild ungulates often forage and damage cropland, as well as compete for resources with livestock, potentially creating sources of conflicts between humans and wild ungulates [14,15]. The widespread reduction in apex predators and more restricted hunting management has contributed to an increase in the abundance of wild ungulates, sometimes resulting in an intensifying interspecific competition pressure [16,17]. The extraordinary population increase has brought new challenges for conservation and management [17,18], yet little is known about the consequence of wild ungulate population explosions in their ecological communities in some areas. Understanding ecological roles and behaviors of wild ungulates, under increasing population densities, is particularly helpful for conservation and management of wild ungulate with growing intensities of human-ungulate conflict [19,20].

Wild ungulates generally show high species richness and abundance, and exhibit greater diverse resources and habitat use. The behavioral studies of wild ungulates have included home ranges, activity patterns, seasonal migrations and groups, foraging and licking salt [21,22,23,24,25,26,27,28]. However, most previous behavioral studies of wild ungulates have focused on single species ecology, with fewer studies having addressed the interactions between species. Studies conducted across several seasons to evaluate both spatial and temporal activity patterns of wild ungulates are still scanty [15,29]. Traditional methods depending on direct observations and tagging to survey animal behaviors have limitations (such as complex terrain, dense vegetation and physical capture), impeding the understanding of behavioral ecology [30]. Camera trapping is a non-invasive method where cameras are left unattended in the field for several months, making the study of wild ungulates spatial coexistence and temporal activity patterns more feasible [31]. Furthermore, animal behaviors recorded by remote cameras are typically a cumulative composition of many individuals, allowing for population-level analyses [32]. Camera traps have grown in popularity among researchers investigating activity patterns with adequate behavioral information captured by remote cameras [33]. A number of recent studies have demonstrated the utility of this technique to quantify activity patterns of target species and their interspecific overlaps [34,35,36,37,38,39,40].

The Qinling Mountains of central China are a biodiversity hotspot and a wildlife-rich area in the world [41]. The area harbors seven wild ungulates, including forest musk deer (*Moschus berezovskii*), which are listed as Endangered by the IUCN red list; golden takin (*Budorcas taxicolor bedfordi*), Chinese goral (*Naemorhedus griseus*), and mainland serow (*Capricornis milneedwardsii*), which are listed as Vulnerable; tufted deer (*Elaphodus cephalophus*) listed as Near Threatened, as well as reeves muntjac (*Muntiacus reevesi*) and wild boar (*Sus scrofa*). Currently, wild ungulate populations in the Qinling Mountains are mostly a successful conservation story [42], as their numbers and distribution ranges have increased substantially, resulting in stiff interspecific competition, especially in places where top predators are virtually absent, with the common leopard (*Panthera pardus*) recorded only once in the study area. Given this predisposition to interspecific competition, the spatial segregation and temporal avoidance may be at least partially due to behavioral mechanisms, allowing for coexistence among these sympatric wild ungulates with particular interest. However, the activity patterns of the species in this region are completely unknown due to difficulty in accessing the locations where they occur [43].

The diversity and coexistence of wild ungulate species with similar foraging strategies in forest ecosystems, as well as factors allowing them to coexist, have long fascinated ecologists [43,44]. In this study, an intensive long-term camera trapping effort was made to survey the spatial and temporal coexistence patterns of wild ungulates in Changqing national nature reserve, which is virtually free of top predators. The objectives were as follows: (i) to elucidate spatial distribution patterns of the wild ungulates, (ii) to describe the daily and annual activity patterns, and (iii) to examine the spatial and temporal overlaps among species. The results describe possible co-existence mechanisms among wild ungulate species, which could be used to guide the species management.

## 2. Materials and Methods

### 2.1. Study Area

Changqing National Nature Reserve (Changqing reserve, 107°25′ to 107°45′ E, 33°26′ to 33°43′ N) is located on the southern slopes of the Qinling Mountains in central China (Figure 1). Changqing reserve was established in 1994, added to the IUCN green list of protected areas in 2014, and further upgraded to a national park in 2017. The reserve covers an area of approximately 299 km^2^, with elevations ranging from 800 m (Maoping) to 3040 m (Huorenping Ridege). The average annual temperature is 7 °C and the average annual rainfall is 814 mm. Based on the local climate characteristics, we divided the year into warm season (April–October) and cold season (November–March) [45]. Vegetation varies with elevation: deciduous broadleaf forest is mainly found at lower elevations, mixed broadleaf-conifer forest is found at mid-elevations, and coniferous forest interspersed with subalpine shrubs and meadows is found at higher elevations. The reserve provided protection to the giant panda (*Ailuropoda melanoleuca*), golden monkey (*Rhinopithecus roxellan**a*), and other endangered species [46]. Of all the mammal species present, 2 were evaluated as Endangered by IUCN RedList, 6 as Vulnerable and 4 as Near Threatened [42,47].

### 2.2. Data Collection

We used 100 infrared cameras (Ltl-6210; Shenzhen Ltl Acorn Electronics Co., Ltd., Shenzhen, China) to survey wild ungulates in Changqing national nature reserve from April 2014 to October 2017. The reserve was divided into 4 km^2^ cells (total of 118 cells; cell size 2 km × 2 km, Figure 1) as potential sampling cells. We intended to sample every cell, but harsh terrain allowed us to sample only 90 cells (76% of the 118 cells). To enhance detection probabilities, infrared cameras were set up in areas with maximum wild ungulate signs (e.g., feces and footprints) and along trails with minimum human disturbance within the 2 km × 2 km cells). Using such an approach therefore was biased to such locations. We placed one infrared camera in each cell 4 to 6 months, and then considered whether to relocate to another location within the same cell according to the detections of animals. During any single survey period we positioned the cameras with more than 300 m spacing, promoting spatial independence.

Cameras were mounted on trees at approximately 0.5 m off the ground and set to record time and date when being triggered. Cameras were programmed with moderate sensitive sensor setting, to shoot 2 photos and a 15 s video when being triggered, and time was set to 24 h per day with a 2-min interval between consecutive events. Cameras were maintained in the field for 4 to 6 months, and were inspected for SD cards and batteries upon movement of cameras in cell. No bait was used to attract animals, which is important in situations where the aim of the study is to look at animal behaviors in an unbiased way.

### 2.3. Data Analysis

Photographs and videos were summarized by sites, hour, and date at each camera placement site. To ensure independence of photographic capture events, any consecutive photograph of the same species within 0.5 h was recorded as a single occurrence event, and for all analyses, only independent detections were considered [48]. The number of effective camera trap days was calculated as the time frame between camera setting, and the date of the last photograph or video was taken if a malfunction occurred (based on date stamp). Capture Rate (CR) was used to compare independent detections in different wild ungulate species. Capture Rate for each species in each site was calculated as the number of independent detections for every species, and multiplied by 100 camera trap days [48].

Spatial analysis—Occupancy modelling are the most commonly used approaches to address interspecific competition [49]. These models allowed us to estimate occupancy (*ψ*) and detection (*Pr*) probability for every combination of coexistence ungulates. For each location, we considered whether each ungulate species was detected (One) or non-detection (Zero) during each 30-days period, then created detection matrix using six periods (0–30, 31–60, 61–90, 91–120, 121–150, >150 days). We selected four environment variables to estimate occupancy probability based on previous studies of habitat selection/activity patterns by ungulates [15,43,50], distance to river (Disriv), distance to settlement (Disset), elevation (Ele) and vegetation type (Veg), and used vegetation type and season as covariates for ungulate detection probability analysis. We obtained rivers, settlements and vegetation type data from Changqing administration, elevation was derived from a digital elevation model with a resolution of 30 m from Resource and Environment Science and Data Center (https://www.resdc.cn/Default.aspx (assessed on 20 October 2021)). All environment variables were resampled at a 30 m resolution and put into the same projection using ArcGis 10.8 (ESRI Inc.). Meanwhile, because of ungulates display seasonal activity pattern, we used season as covariate, divided year into two seasons: warm season (from April to October) and cold season (from November to March). We used a logit link function to model occupancy and detection with covariates that varied among camera locations. For each model set, we ranked models based on their Akaike’s Information Criterion (AIC) values, and only the models (ΔAIC ≤ 2) of top model equal-weight average as the optimal model to obtain covariate estimates [51]. Constructions of models were implemented with the “unmarked” package in R.

Spatial overlap—The co-occurrence between wild ungulates was evaluated by applying presence/absence data from the 2 km × 2 km cells through the Sørensen similarity index [52,53]:$$S_{ij}=\frac{2a_{ij}}{2a_{ij}+b_{ij}+c_{ij}}$$

Here a*_ij_* represented the number of 2 km × 2 km cells with the presence of both species *i* and *j*, and b*_ij_* and c*_ij_* were cell numbers with the presence of only one species. This index ranged from 0 (maximum segregation) to 1 (maximum co-occurrence). Spatial overlaps were evaluated, with respect to the overall pairwise comparisons performed, considering *S_ij_* ≤ 0.5 as low overlaps, 0.5 < *S_ij_* ≤ 0.8 as moderate overlaps, and *S_ij_* > 0.8 as high overlaps.

Temporal analysis—Relative activity indices (RAI) were used to estimate annual activity patterns of wild ungulates [54]. The number of independent detections for each camera were summed by each month, and multiplied by 1000 camera trap days.

Independent detections were sorted to examine whether wild ungulates circadian activity patterns were diurnal, nocturnal and crepuscular (1 h before sunrise and 1 h after sunset [35,55]). Since the samplings were carried out throughout the year, we estimated the yearly variations in the sunrise and sunset times from the April 2014 to October 2017. Average sunrise and sunset time were at 06:37 and 19:01 in the study area. Selection ratios were calculated, of each time period for each species with the following calculation [56,57]:$$w_{i}=o_{i}/\pi_{i}$$

Here *w_i_* was the selection ratio for period *i*; o*_i_* was the proportion of detections in period *i*; and π*_i_* was the proportion of length in period *i* to the length of all periods. *w_i_* > 1 indicated that the time period was selectively used; *w_i_* < 1 indicated the time period was avoided.

Daily temporal overlaps—Activity time data of each species were transformed from hours to degrees. Pairwise comparisons of wild ungulate activity patterns were performed by estimating the coefficients of overlaps (Δ), and the estimator Δ_4_ was used whenever the smallest sample in the dataset was >75 records, otherwise Δ_1_ was used [58]. The coefficients of overlaps ranged from 0 (no overlap) to 1 (complete overlap), and were obtained by taking the minimum of the density functions of two cycles being compared at each time point [59]. The precision of this estimator was obtained by computing a standard deviation from 10,000 bootstrapping samples. These activity pattern analyses were performed by using “overlap” package for R [60,61]. Activity overlap values were evaluated, with respect to the overall pairwise comparisons performed, considering Δ value ≤ 0.5 as low overlaps, 0.5 < Δ ≤ 0.75 as moderate overlaps, and Δ > 0.75 as high overlaps [62].

Kruskal-Wallis was used to compare elevational and RAI differences of sites among wild ungulates, followed by a Post Hoc Test (Duncan’s Multiple Range Comparison Test) to determine the significance. Kruskal-Wallis was used to determine if the species used these three time periods differently (diurnal, crepuscular and nocturnal). Data were expressed as Mean ± Standard deviation (Mean ± SD). Statistical significance was set at *p* < 0.05.

## 3. Results

### 3.1. General Summary

A total of 620 sites were surveyed, of which 47 failed due to camera damage, malfunction, or theft, resulting in 573 sites for data analysis. We obtained 15,584 independent detections of seven wild ungulate species (out of 38,805 total detections) during 93,606 camera-trap days (number of camera-trap days in each site was 162 ± 51 days) from April 2014 to October 2017, and seven wild ungulate species were detected in the study area (Table 1). The capture rate (CR) per 100 camera trap days differed significantly across species (χ^2^ = 262.66, *df* = 6, *p* < 0.001; Table 1).

The CR values of wild ungulate species ranked from the highest to the lowest as follows: reeves muntjac, wild boar, golden takin, tufted deer, Chinese goral, forest musk deer and mainland serow. Duncan’s multiple range test indicated that the CR values of reeves muntjac showed a significant difference from other wild ungulates, but wild boar and golden takin showed no significant difference, and neither did the pairwise of forest musk deer, tufted deer, mainland serow and Chinese goral.

### 3.2. Spatial Utilization Distribution

The wild boar had higher occupancy rates than other species based on a top model (ΔAIC ≤ 2; Table 2), with average occupancy rate estimated as 0.888 (*ψ*), followed by golden takin (*ψ* = 0.712), tufted deer (*ψ* = 0.553), muntjac deer (*ψ* = 0.486), Chinese goral (*ψ* = 0.318), mainland serow (*ψ* = 0.226) and forest musk deer (*ψ* = 0.460). Vegetation type had a positive relationship with forest musk deer (*β* = 0.62 ± 0.55; Table 3), muntjac deer (*β* = 0.69 ± 0.11), mainland serow (*β* = 0.61 ± 0.51), Chinese goral (*β* = 0.11 ± 0.14), and negative with golden takin (*β* = −1.05 ± 0.13). Distance to settlements had a negative relationship with wild boar (*β* = −0.24 ± 0.17), muntjac deer (*β* = −0.15 ± 0.06) and mainland serow (*β* = −0.33 ± 0.29). Elevation had position relationship with forest musk deer (*β* = 0.66 ± 0.23) and golden takin (*β* = 1.22 ± 0.17), and negative relationship with muntjac deer (*β* = −0.34 ± 0.24). Spatial projection of ungulates occupancy in Changqing reserve shown in Figure 2.

The detection probability was different among ungulates (Table 2), and was highest for wild boar (*Pr* = 0.599) and lowest for mainland serow (*Pr* = 0.080). Vegetation type had a positive relationship with detection probability of wild boar (*β* = 0.28 ± 0.04; Table 3), forest musk deer (*β* = 0.21 ± 0.53), tufted deer (*β* = 0.19 ± 0.07), muntjac deer (*β* = 0.15 ± 0.06), Chinese goral (*β* = 0.26 ± 0.12), and negative with mainland serow (*β* = −0.34 ± 0.12). Season had a negative relationship with detection probability of wild boar (*β* = −0.08 ± 0.04), tufted deer (*β* = −0.76 ± 0.08), mainland serow (*β* = −0.10 ± 0.17), and Chinese goral (*β* = −0.34 ± 0.11).

### 3.3. Spatial Overlaps

Forest musk deer and mainland serow had relatively low spatial overlaps with other wild ungulates (*S_ij_* ≤ 0.5; Table 4), and forest musk deer and mainland serow had the lowest spatial overlaps (*S_ij_* = 0.26). Chinese goral had moderate spatial overlaps with reeves muntjac (0.5 < *S_ij_* ≤ 0.75), and relatively high spatial overlaps with golden takin (*S_ij_* = 0.81), wild boar (*S_ij_* = 0.77) and tufted deer (*S_ij_* = 0.76). Spatial overlaps among wild boar, tufted deer, reeves muntjac and golden takin were relatively high (*S_ij_* > 0.8), and wild boar and golden takin had almost perfect spatial overlaps (*S_ij_* = 0.96).

### 3.4. Annual Activity Patterns

The mean survey efforts in each month were 7832 ± 1212 camera-days from April 2014 to October 2017. Maximum effort was 9111 camera-days in May, and minimum effort was 6099 camera-days in February. Based on relative activity indices (RAI), wild boar showed higher activity between August and October with a peak in September (Figure 3), and lower activity from February to April. Tufted deer showed higher activity between May and September with a peak in June and lower activity from October to April. Both reeves muntjac and Chinese goral showed higher activity in July, and lower activity from January to April. Mainland serow was more active in May than other months, and was not detected in March and October. Forest musk deer showed more activity in December, and lower activity from August to October and February to April. Golden takin displayed two high annual periods, the first being in April and the second from September to November, with RAI during these months higher than in other months.

### 3.5. Daily Temporal Patterns

Based on selection ratios, all wild ungulate species (expect wild boar) were mainly crepuscular and/or diurnal (*w_i_* > 1; Table 5), and showed bimodal activity peaks at around 05:00–07:00 and 17:00–19:00, with low levels of activity during nocturnal time period (*w_i_* ≤ 1). Wild boar were more diurnal (*w_i_* = 1.71) than crepuscular (*w_i_* = 0.81) and nocturnal (*w_i_* = 0.29), showed single daily activity peak at around 10:00–14:00.

### 3.6. Temporal Overlaps

Forest musk deer had moderate temporal overlaps with other five wild ungulate species (0.5 < Δ ≤ 0.75; Table 4), and forest musk deer and wild boar had the lowest temporal overlaps (Δ = 0.54). Temporal overlaps among wild boar, tufted deer, reeves muntjac, golden takin, mainland serow and Chinese goral were relatively high (all pairwise Δ > 0.75), and that among tufted deer and reeves muntjac (Δ = 0.9), and reeves muntjac and Chinese goral (Δ = 0.9) had the highest temporal overlaps (Figure 4).

## 4. Discussion

This intensive camera-trap survey elucidated much more detailed studies of spatial and temporal activity patterns among multiple sympatric wild ungulates under natural conditions, which could be useful to conservation and management of wildlife. Camera trapping provided new analytical methods and insights, enabling scientists to quantify behaviors of wild ungulates, which was robust to variation in field conditions and allowed for the collection of data on multiple species with less time, labor and disturbance to wildlife [63]. Moreover, camera trapping provided detailed biological information, allowing us to gain important insights into potential interspecific interactions of sympatric species.

The results showed significant differences in peaks of annual activity patterns of wild ungulates. Such patterns may link the abundance and quality of food resources in different seasons, and, furthermore, temporal adjustment can minimize competition among sympatric wild ungulates [15,43,64]. The relative activity indices of wild boar, tufted deer, reeves muntjac and Chinese goral exhibited significantly higher activities during the warm season (mainly from May to September), similar to previous studies [15,65,66,67,68]. Interestingly, the highest activities of wild boar were in September, tufted deer in June, and both reeves muntjac and Chinese goral in July. It was found that activity levels of wild boar, tufted deer, reeves muntjac and Chinese goral during winter were lower than that measured during summer, for the reduction of activity levels and movements meant to likely conserve energy, and spend more time to digest low-quality food [34]. In contrast, golden takins showed relatively lower activities in the warm season and were more active in spring and autumn, exhibiting a large degree of temporal separation in their annual activity patterns with the above four species. This coincided with previous studies, which found the annual activity patterns of golden takins were consistent with seasonal altitudinal migration (spring and autumn) and reproductive periods (most golden takins were in larger groups at high altitudes and engaging in rutting behavior from June to August) [15]. Detection rates of forest musk deer and mainland serow were relatively low, yet the annual activity patterns of forest musk deer were similar to that of a previous study, which also recorded more activities of species in December and lower activities in August [43]. The underlying mechanisms of annual activity patterns of forest musk deer and mainland serow were not yet fully understood and required further study. In particular, no detection of mainland serow was recorded in March and October, even with a four-year survey effort.

Daily temporal avoidance could reduce competition and thus facilitate species coexistence in cases where species segregated between diurnal, crepuscular, and nocturnal domains [69]. Based on the results, only moderate temporal avoidance between forest musk deer and other wild ungulates was observed daily. The remaining pairs of wild ungulates exhibited higher degrees of temporal overlaps in their daily activity patterns, suggesting that daily temporal pattern was not a major factor contributing to their coexistence in the area. All wild ungulates (except wild boar) were mainly active during crepuscular and diurnal, and showed bimodal activity peaks at crepuscular. Such a bimodal pattern with prominent peaks around sunrise and sunset had been found for most wild ungulates [39,43,70,71,72,73]. Mainland serow showed a high level of activities during diurnal and crepuscular periods, which was contrast to that of another published work [43], which found mainland serow to be nocturnal. Even though daily activity intensity of forest musk deer was greater during the diurnal (*w_i_* = 1.16), they still exhibited clear crepuscular activities (*w_i_* = 1.68), which was similar to that of a previous study [65]. Solar radiation was likely the main environmental factor to explain daily activity patterns of wild ungulates [34]. Generally, most wild ungulates’ daily activity peaks occurred at crepuscular when temperatures were relatively cool with low humidity, and these periods were spent foraging and moving slowly [70]. Peaks in resting behaviors often occurred after foraging behaviors and some (e.g., golden takin) were often accompanied with rumination. Ungulates may lay under trees or stone cliffs to cool their bodies during the hottest part of the day, reducing their overall movement and energy expenditure during these periods, especially in mid-summer [15]. Interestingly, it also was found a small number of golden takins were still moving or engaged in other activities (foraging or mating) later in the night (around 23:30–02:00), which could recommend that immediate studies be undertaken to investigate the cause of such changes.

The spatial distribution patterns of wildlife population and communities attracted broad interests in ecology research. However, knowledge on the patterns of wild ungulates is still poorly understood [44]. The population of mainland serow and forest musk deer is rare and elusive, and both show high spatial avoidance with other species. Vertical spatial distribution may be the most important mechanism of coexistence. Forest musk deer only concentrated in high-elevation zones, with coniferous forest interspersing with some subalpine shrubs and meadows at these zones. Mainland serow were mainly distributed in low-elevation zones, with deciduous broadleaf forest at these zones, where the understory were dominated shrubs containing adequate food resources. Golden takins exhibited distinct elevation seasonal migration [15,50], with migration patterns from high-elevation meadows in summer to mid-elevation fir forest and bamboo in winter, and low-elevation valleys in spring and autumn, likely as a result of their relatively high spatial overlaps with other wild ungulates. For pairs of wild boar, tufted deer and reeves muntjac, high temporal and spatial overlaps were observed with no distinct elevational differences. Our results may imply the presence of other niche dimensions that were not considered in our study, such as diet. Previous studies on the diet of these three species showed that tufted deer mainly consumed green parts of the vegetation (tender twigs and leaves) [74,75], while reeves muntjac fed on tender twigs and leaves as well as seeds, fruits and fungi [76]; wild boar was a generalist and opportunistic species, whose diet composition was extremely plastic (such as tuber, fruits, crops, rodents, and earthworms), and therefore it was able to adapt well to the consumption of the food sources available under various habitat conditions [77].

The success of giant panda conservation in the study area appeared to have benefited sympatric wild ungulate species. For example, with the exception of the wild boar, previous studies have documented that wild ungulates were rarely found below 1360 m in study area (ecotone of forestry and agriculture [44,50]). However, our camera traps detected most golden takins, mainland serow, reeves muntjac, tufted deer and Chinese goral at lower elevation zones (less than 1360 m). This change may be due to the implementation of conservation programs, having returned low-elevation farmland to forests. Over the past few decades, the Chinese government had implemented numerous conservation programs, including the establishment of nature reserves, Grain-to-Green program and the Natural Forest Conservation Program, to protect and improve habitats for native wildlife [78,79]. Most of the known key threats to the species were being mitigated, and most wild ungulate populations were rapidly recovering. Dramatic increases in populations of wild ungulates brought a new ecological issue for wildlife ecologists and managers [18,80]. Actually, not only were wild boars easy to see in the Qinling Mountains, but golden takins and reeves muntjac had significantly increased their distributions, and it was possible to see tufted deer frequently near villages. Wild ungulates foraged and damaged considerable agricultural crops around the reserve, even hurt humans [81,82], impacting the enthusiasm of local villagers involved in the national park. Consequently, it was recommended that the populations of wild ungulates should be monitored, and compensation for local villagers’ loss caused by wild ungulates must be taken into consideration. In addition, it was necessary to understand the species’ ecological and behavior requirements if wild ungulate corridors were going to be established through altered habitats and agricultural land.

Carnivores played important roles in structuring communities [83], as the removal of top predators may contribute to the population explosion of ungulates. The growth or expansion of ungulate species may have unexpected impact on previously established populations of coexistence, as well as a variety of effect on forest ecosystems, many of which are not fully understood. China launched an ambitious program to establish a national park system, integrating current protected areas to resolve the problems of fragmented management [84]. This park provided a unique opportunity to coordinate strategies and polices to restore large carnivore populations at local and landscape levels [85]. However, it is not fully understood how ungulates would respond to current strategies, and what management measures would be more effective to facilitate the recovery of large carnivore populations. Understanding the ecology of ungulate prey was important to predator conservation [86]. Thus, long-term standardized monitoring programs on ungulates and their habitats will be necessary. Such programs could provide evidence-based insight into the direct impacts of future large predator restoration (although this might require decades for goal fulfillment). In addition, tolerance among local people and engaging policy makers to support the establishment of national parks along with targeted conservation monitoring was critical, so as to ensure that wildlife and people could co-exist in this area.

The study had some limitations. First, some locations were unable to be sampled due to the difficulty of navigating in remote areas, which may result in detection errors or bias, whereby wild ungulates may have been present at these sites regardless of the ability to survey them. However, the field survey still covered 76% of the study area, a sizable portion of the intended land mass, and it was observed that temporal activities were consistent with most previous studies. Second, illustrating imperfect detection in camera trapping surveys of unmarked species was difficult, and distinguishing between individuals, sex and age classes from camera trapping photographs was challenge, because the study did not evaluate environmental factors (e.g., human disturbances) that affected the activity patterns of these wild ungulates. Further, detection probability may be influenced by relocating cameras to another location within the same cell. Thus, we analyzed a subset of the data (from January to December 2016). Temporal and spatial activity patterns of sympatric wild ungulates showed similar tends to that of the overall dataset (from April 2014 to October 2017). Because of the limitations of camera-trapping surveys, future studies are required to clarify various influences on the activity patterns of wild ungulates, in combination with other survey tools such as telemetry surveys and direct observations.

## 5. Conclusions

Our study provides novel information on the spatial and temporal ecology of sympatric wild ungulates in forest ecosystems in a scenario of potential disturbance caused by the reduction of apex predators. This information is important to set a baseline of understanding of mechanisms of ecological interaction among wild ungulates, from which we can interpret any changes in ungulates abundance and ecology, and further consequent trophic cascades, with the progressive and population explosions of wild ungulates. Furthermore, the results could be used as assessment of wild ungulate conservation status in the study area, which could be a guide to establishing conservation priorities as well as efficient management programs.

## Figures and Tables

**Figure 1 animals-12-01666-f001:**
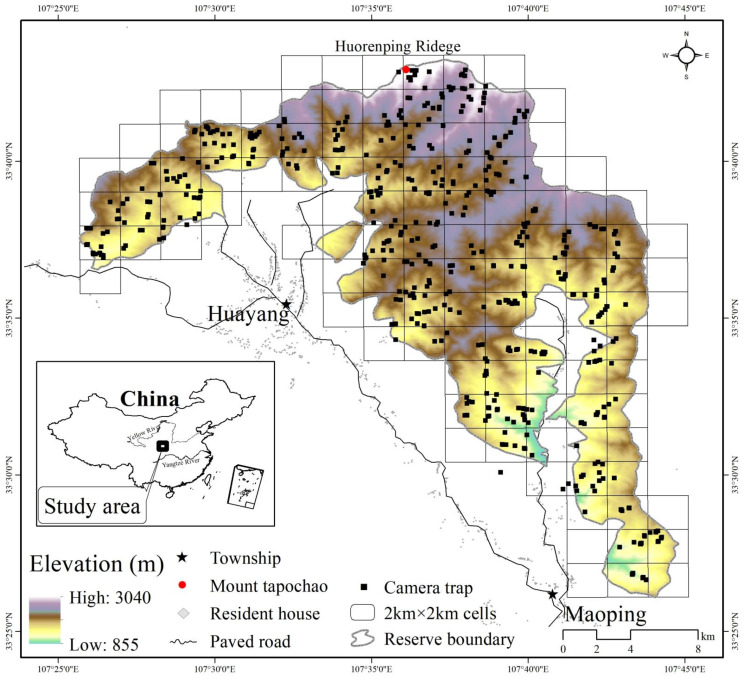
Locations of camera trapping stations (black square) distributed in 2 km × 2 km cells in the Changqing national nature reserve. Inset map shows the study area location in central China. Elevation ranges from approximately 800 m to 3040 m.

**Figure 2 animals-12-01666-f002:**
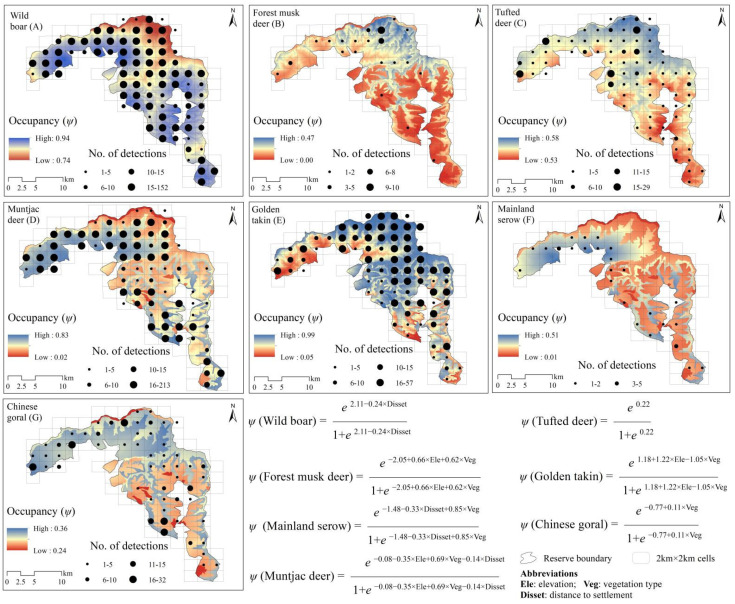
Spatial projection of ungulates occupancy probability (*ψ*) and model structure in Changqing reserve, based on the average top models (∆AIC ≤ 2). (**A**)—wild boar, (**B**)—forest musk deer, (**C**)—tufted deer, (**D**)—muntjac deer, (**E**)—golden takin, (**F**)—mainland serow, (**G**)—Chinese goral. Black points show the center of each 2 km × 2 km cell, and their size indicates the total number of independent camera detections.

**Figure 3 animals-12-01666-f003:**
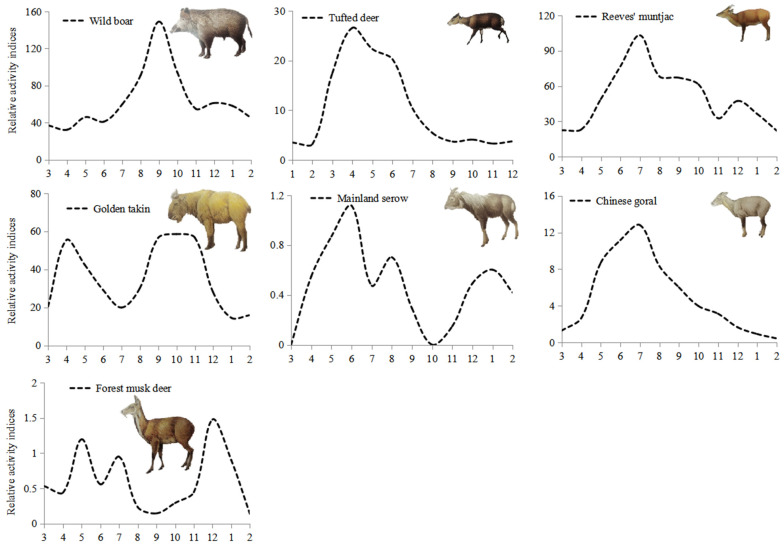
Relative activity indices (*RAI*, number of detections per 1000 camera-days) for each month and the number of total detections (*n*) for each wild ungulate species in the Changqing National Nature Reserve, China.

**Figure 4 animals-12-01666-f004:**
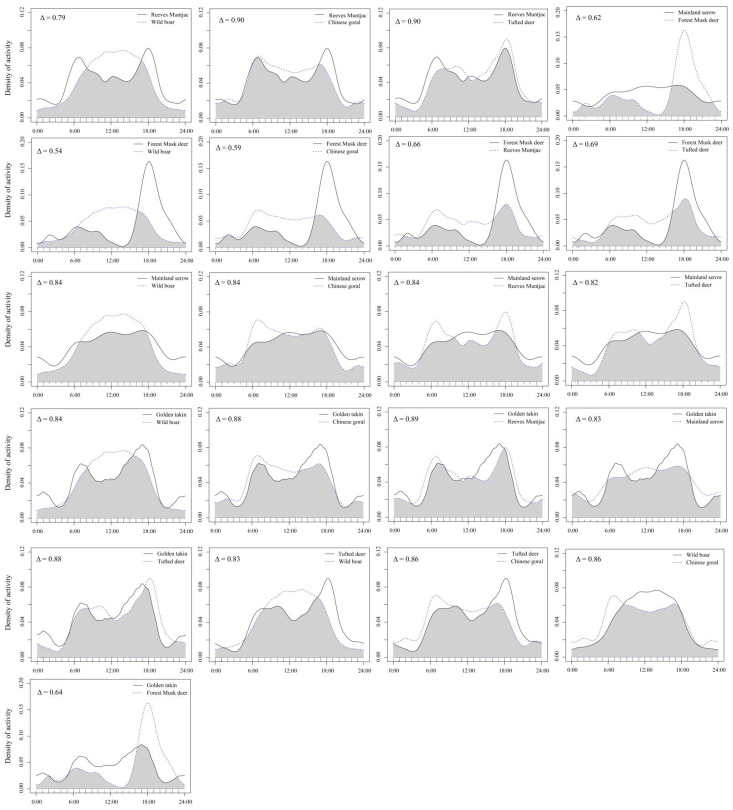
Daily activity patterns and temporal overlapped different pairs of wild ungulate species in the Changqing National Nature Reserve, China. Overlaps are represented by the shaded gray area. The *y*-axis is the “Kernel Density Estimates”.

**Table 1 animals-12-01666-t001:** The number of sites, 2 km × 2 km cells and independent detections, and capture rates of wild ungulates in the Changqing National Nature Reserve. Duncan’s multiple range comparison test results of differences among species are shown at their respective columns. Means with varying superscript letters indicate significant differences (*p* < 0.05).

Species	No. of Sites	No. of 2 km × 2 km Cells	No. of Independent Detections (*n*)	Capture Rate per 1000 Camera Trap Days (Mean ± SD)
Wild boar (*Sus scrofa*)	495	87	5827	(7.05 ± 8.52) ^a^
Forest musk deer (*Moschus berezovskii*)	33	21	57	(1.12 ± 1.38) ^b^
Tufted deer (*Elaphodus cephalophus*)	266	76	1029	(2.74 ± 3.60) ^b^
Reeves’ muntjac (*Muntiacus reevesi*)	275	75	4799	(10.01 ± 14.17) ^c^
Golden takin (*Budorcas taxicolor bedfordi*)	382	81	3323	(5.24 ± 6.47) ^a^
Mainland serow (*Capricornis milneedwardsii*)	33	46	46	(1.00 ± 1.07) ^b^
Chinese goral (*Naemorhedus griseus*)	151	55	503	(2.29 ± 2.76) ^b^

**Table 2 animals-12-01666-t002:** Summary of model selection results for ungulate occupancy in the Changqing reserve, showing estimated occupancy rate (*ψ*) and detection probability (*Pr*) for the average top models (∆AIC ≤ 2). Abbreviations: Veg-vegetation type, Disset-distance to settlement, Disriv-distance to rivers, Ele-elevation.

Species	Models	Number of Parameters	AIC	ΔAIC	AICWt	*ψ*	*Pr*
Wild boar	psi (Disset); p (Season + Veg)	5	3919.84	0.00	0.14	0.889	0.599
	psi (Disset + Veg); p (Season + Veg)	6	3919.85	0.10	0.14	0.887	0.600
	psi (Disset + Disriv); p (Season + Veg)	6	3920.21	0.37	0.12	0.888	0.599
	psi (Ele + Veg); p (Season + Veg)	6	3921.76	1.92	0.05	0.887	0.600
	psi (Ele); p (Season + Veg)	5	3921.79	1.95	0.05	0.889	0.599
	psi (Ele + Disset); p (Season + Veg)	6	3921.83	1.99	0.05	0.889	0.600
	**Model average**					**0.888**	**0.599**
Forest musk deer	psi (Ele + Veg); p (Veg)	5	437.24	0.00	0.20	0.141	0.114
	psi (Ele + Veg); p (Season)	5	437.29	0.04	0.20	0.140	0.107
	psi (Ele); p (Veg)	4	437.76	0.52	0.16	0.190	0.075
	psi (Ele + Disriv); p (Veg)	5	439.03	1.79	0.08	0.186	0.077
	psi (Ele + Veg); p (Season + Veg)	6	439.24	1.99	0.07	0.141	0.115
	**Model average**					**0.160**	**0.098**
Tufted deer	psi (.); p (Season + Veg)	4	2302.17	0.00	0.23	0.554	0.277
	psi (Ele); p (Season + Veg)	5	2303.45	1.27	0.12	0.553	0.277
	psi(Veg);p (Season + Veg)	5	2303.62	1.45	0.11	0.552	0.278
	psi (Disriv); p (Season + Veg)	5	2303.65	1.48	0.11	0.553	0.277
	psi (Disset); p (Season + Veg)	5	2303.04	1.86	0.09	0.554	0.277
	**Model average**					**0.553**	**0.277**
Muntjac deer	psi (Ele + Veg); p (Veg)	5	2703.02	0.00	0.41	0.486	0.584
	psi (Disset + Veg); p (Veg)	5	2704.15	1.13	0.24	0.485	0.584
	psi (Ele + Veg); p (Season + Veg)	6	2705.01	1.99	0.15	0.486	0.584
	**Model average**					**0.486**	**0.584**
Golden takin	psi (Ele + Veg); p (Veg)	5	3487.72	0.00	0.17	0.713	0.440
	psi (Ele); p (Veg)	4	3488.39	0.67	0.12	0.712	0.441
	psi (Ele + Veg); p (Season + Veg)	6	3488.76	1.03	0.10	0.714	0.439
	psi (Ele); p (.)	3	3489	1.28	0.09	0.710	0.442
	psi (Ele + Veg); p (.)	4	3489.27	1.55	0.08	0.712	0.442
	psi (Ele); p (Season + Veg)	5	3489.36	1.64	0.07	0.711	0.440
	**Model average**					**0.712**	**0.441**
Mainland serow	psi (Disset + Veg); p (.)	4	431.6	0.00	0.13	0.201	0.068
	psi (Disset + Veg); p (Veg)	5	431.82	0.22	0.12	0.217	0.090
	psi (Disset + Veg); p (Season + Veg)	6	432.06	0.46	0.11	0.226	0.092
	psi (Disset + Veg); p (Season)	5	432.34	0.74	0.09	0.208	0.066
	psi (Ele + Veg); p (Season + Veg)	6	433.02	1.42	0.07	0.268	0.089
	psi (Ele + Veg); p (.)	4	433.43	1.83	0.05	0.208	0.066
	psi (Veg); p (Veg)	4	433.47	1.87	0.05	0.254	0.086
	**Model average**					**0.226**	**0.080**
Chinese goral	psi (Veg); p (Season + Veg)	5	1500.53	0.00	0.19	0.322	0.254
	psi (.); p (Season + Veg)	4	1501.3	0.77	0.13	0.311	0.262
	psi (Veg); p (Season)	4	1501.72	1.19	0.10	0.322	0.254
	psi (Disset + Veg); p (Season + Veg)	6	1502.3	1.77	0.08	0.311	0.262
	psi (Ele + Veg); p (Season + Veg)	6	1502.52	1.99	0.07	0.322	0.254
	psi (Disriv + Veg); p (Season + Veg)	6	1502.53	1.99	0.07	0.323	0.254
	**Model average**					**0.318**	**0.257**

**Table 3 animals-12-01666-t003:** Covariates influencing ungulates occupancy and detection probability according to coefficients and (*β*) associated standard errors (SE). Abbreviations: Veg-vegetation type, Disset-distance to settlement, Disriv-distance to rivers, Ele-elevation.

Species	Model Component	Covariates	Estimate (*β*)	SE	*Z*	*p*
Wild boar	Occupancy	Intercept	2.11	0.15	13.86	<0.001 ***
		Disset	−0.24	0.17	1.36	0.174
		Veg	0.07	0.13	0.55	0.581
		Disriv	−0.04	0.10	0.39	0.698
		Ele	−0.04	0.13	0.33	0.740
	Detection	Intercept	0.41	0.04	9.97	<0.001 ***
		Veg	0.28	0.04	6.70	<0.001 ***
		Season	−0.08	0.04	2.02	0.0436*
Forest musk deer	Occupancy	Intercept	−2.05	0.50	4.12	<0.001 ***
		Ele	0.66	0.23	2.89	0.003 **
		Veg	0.62	0.55	1.12	0.263
		Disriv	−0.02	0.10	0.22	0.823
	Detection	Intercept	−2.30	0.49	4.72	<0.001 ***
		Veg	0.21	0.53	0.40	0.689
		Season	0.00	0.12	0.03	0.973
Tufted deer	Occupancy	Intercept	0.22	0.13	1.70	0.090
		Ele	0.02	0.06	0.29	0.775
		Veg	0.01	0.06	0.25	0.808
		Disriv	−0.01	0.05	0.25	0.807
		Disset	0.01	0.04	0.13	0.898
	Detection	Intercept	−1.09	0.08	12.12	<0.001 ***
		Season	−0.76	0.08	9.42	<0.001 ***
		Veg	0.19	0.07	2.93	0.003 **
Muntjac deer	Occupancy	Intercept	−0.08	0.09	0.85	0.40
		Ele	−0.35	0.24	1.45	0.15
		Veg	0.69	0.11	6.56	<0.001 ***
		Disset	−0.14	0.22	0.63	0.53
	Detection	Intercept	0.34	0.06	5.95	<0.001 ***
		Veg	0.15	0.06	2.48	0.013 *
		Season	0.00	0.02	0.04	0.969
Golden takin	Occupancy	Intercept	1.18	0.14	8.60	<0.001 ***
		Ele	1.22	0.17	7.18	<0.001 ***
		Veg	−1.05	0.13	0.78	0.436
	Detection	Intercept	−0.24	0.05	5.04	<0.001 ***
		Veg	0.06	0.05	1.10	0.272
		Season	0.01	0.03	0.40	0.692
Mainland serow	Occupancy	Intercept	−1.48	0.45	3.25	0.001 **
		Disset	−0.33	0.29	1.15	0.252
		Veg	0.85	0.51	1.66	0.096
		Ele	−0.07	0.18	0.39	0.690
	Detection	Intercept	−2.56	0.39	6.64	<0.001 ***
		Veg	−0.34	0.42	0.81	0.420
		Season	−0.10	0.17	0.59	0.558
Chinese goral	Occupancy	Intercept	−0.77	0.13	5.84	<0.001 ***
		Veg	0.11	0.14	0.76	0.450
		Disset	0.04	0.10	0.42	0.677
		Ele	−0.01	0.09	0.07	0.946
		Disriv	0.00	0.03	0.03	0.980
	Detection	Intercept	−1.10	0.13	8.75	<0.001 ***
		Season	−0.34	0.11	3.07	0.021 *
		Veg	0.26	0.12	2.13	0.035 *

Note: the different superscript letters represent the significant, *** *p* < 0.001, ** 0.001 < *p* < 0.01, * 0.01 < *p* < 0.05.

**Table 4 animals-12-01666-t004:** The spatial overlap index (*S_ij_*), diel activity overlap (Δ) and confidence intervals of each species pair among the wild ungulates detected in the Changqing National Nature Reserve, China.

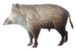 Wild boar	Δ = 0.54 (0.43–0.63)	Δ = 0.83 (0.79–0.84)	Δ = 0.79 (0.77–0.80)	Δ = 0.84 (0.81–0.84)	Δ = 0.84 (0.72–0.91)	Δ = 0.86 (0.82–0.89)
*S_ij_* = 0.34	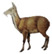 Forest musk deer	Δ = 0.69 (0.57–0.78)	Δ = 0.66 (0.54–0.76)	Δ = 0.64 (0.52–0.73)	Δ = 0.62 (0.46–0.76)	Δ = 0.59 (0.46–0.69)
*S_ij_* = 0.93	*S_ij_* = 0.43	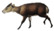 Tufted deer	Δ = 0.90 (0.87–0.92)	Δ = 0.88 (0.84–0.90)	Δ = 0.82 (0.80–0.94)	Δ = 0.86 (0.82–0.90)
*S_ij_* = 0.93	*S_ij_* = 0.42	*Sij* = 0.90	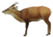 Reeves muntjac	Δ = 0.89 (0.87–0.91)	Δ = 0.84 (0.81–0.94)	Δ = 0.90 (0.87–0.93)
*S_ij_* = 0.96	*S_ij_* = 0.41	*S_ij_* = 0.91	*S_ij_* = 0.88	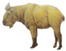 Golden takin	Δ = 0.83 (0.81–0.93)	Δ = 0.88 (0.85–0.92)
*S_ij_* = 0.44	*S_ij_* = 0.26	*S_ij_* = 0.46	*S_ij_* = 0.46	*S_ij_* = 0.45	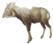 Mainland serow	Δ = 0.84 (0.83–0.95)
*S_ij_* = 0.77	*S_ij_* = 0.39	*S_ij_* = 0.76	*S_ij_* = 0.74	*S_ij_* = 0.81	*S_ij_* = 0.40	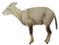 Chinese goral

Note: the shade of grey represents the degree of overlap.

**Table 5 animals-12-01666-t005:** The number of independent detections *n* (selection ratio: *w_i_*) and random use test results of diurnal, nocturnal, and crepuscular (1 h before sunrise and 1 h after sunset, and Average sunrise and sunset time were at 06:37 and 19:01) time periods for wild ungulates in the Changqing National Nature Reserve, China.

Species	*n*(*w_i_*) in Time Period			Kruskal-Wallis Tests (χ^2^, *df* = 2)
	Diurnal	Nocturnal	Crepuscular	
Wild boar (*Sus scrofa*)	4368(1.71)	668(0.29)	791(0.81)	360.99, *p* < 0.001
Forest musk deer (*Moschus berezovskii*)	29(1.16)	12(0.53)	16(1.68)	3.44, *p* = 0.179
Tufted deer (*Elaphodus cephalophus*)	661(1.47)	138(0.34)	230(1.34)	4.56, *p =* 0.102
Reeves’ muntjac (*Muntiacus reevesi*)	2747(1.31)	911(0.48)	1141(1.43)	509.63, *p* < 0.001
Golden takin (*Budorcas* *taxicolor bedfordi*)	1992(1.37)	624(0.47)	707(1.28)	198.64, *p* < 0.001
Mainland serow (*Capricornis milneedwardsii*)	27(1.34)	8(0.44)	11(1.43)	4.98, *p* = 0.083
Chinese goral (*Naemorhedus griseus*)	315(1.43)	83(0.42)	105(1.25)	72.69, *p* < 0.001

## Data Availability

All data during our study was supplied regarding data availability: Raw data is available in the Appendix A. Camera trapping detections of wild ungulates and survey effort of each site in Changqing National Nature Reserve, China Appendix A included the filtrate information for wild ungulates in each camera survey from April 2014 to October 2017, Photograph name (column B), Site ID (column C), Photograph data and time (column D and E), Species type, name and population in each photograph (column F, G and H), GPS information (column I, J and K), the date the placed in operation and the date the last photographs was taken if a malfunction had occurred (column L and M). The photographs captured by the infrared camera collected from each site are all put into a folder named after the camera placement date and camera number, such as the folder name 20140417-01, 20140417 represent the placement date, 01 represent the camera number. Then, rename each photograph in each folder so that each photograph has a unique serial number to distinguish them from each other, such as CQ-20140417-01-0009, CQ is short for Changqing, 0009 represent the photograph name.

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
