# Peer review of "Temporal and Spatial Activity Patterns of Sympatric Wild Ungulates in Qinling Mountains, China"

_animals, 2022, doi:10.3390/ani12131666_

Round 1
Reviewer 1 Report
This article reported the present distribution and activity pattern for the 7 co-existing
wild ungulate species in Qinling Mountains, central China. The amount of data
collected by the camera trappers in this study was sufficient for the above-
mentioned purposes, as mentioned in the Abstract that “The results provided
detailed information of the spatial and temporal ecology of wild ungulate
communities in forest ecosystems of China (Line 42-43)”. However, this article
requires major revision before it can be considered for publication and the following
issues should be addressed:
1. the important application of the information generated from this study for “be a
guide to establish conservation priorities as well as efficient management
programs. (Line 43-44)” was only poorly addressed in the discussion (Line 462-
467) and need to be improved substantially. For example, what management
strategy should be adapted for the expansion of ungulate populations in the
Qinling Mountains? Should the expansion be encouraged by habitat management
(e.g., the establishment of ungulate corridors)? Or their expansions should be
suppressed to reduce the possible impacts to the local ecosystem and
surrounding farmlands? And how to do it based on the evidence (e.g., their
spatial distribution patterns) provided by this study, any suggestions?
2. the discussion on the mechanisms of coexistence (i.e., interspecific competition
among ungulate species), as a theme of this article, was an over-interpretation of
the data. Implication of the existing of interspecific competition requires at least
evidence of the differences between the historical and present patterns in spatial
and temperate distribution, or, better, differences of these distribution patterns
between the studied population and the other populations with different
community structure. There were no evidence supports this argument in this
article, my suggestion is to remove all arguments concerning competition from
this article.
3. Line 369-372: the attribution of the differences in annual activity patterns of
sympatric ungulate species to minimize competition require solid evidence or
references.
4. Line 61-70 in the Introduction is not relevant to this study.
5. Line 444-449: hard to understand the meaning of this sentence
Author Response
Dear Reviewer:
We would like to thank Animals for giving us the opportunity to revise our manuscript again. Thank you for your comments on our manuscript. Those comments are very helpful for revising and improving our paper. We read all the comments carefully and tried our best to revise the manuscript, and have made much correction which we hope meet with approval. Below is our response to your comments.
Thanks for all the help
Best wishes,
Li Jia
#Reviewer
This article reported the present distribution and activity pattern for the 7 co-existing wild ungulate species in Qinling Mountains, central China. The amount of data collected by the camera trappers in this study was sufficient for the above mentioned purposes, as mentioned in the Abstract that “The results provided detailed information of the spatial and temporal ecology of wild ungulate communities in forest ecosystems of China (Line 42-43)”. However, this article requires major revision before it can be considered for publication and the following issues should be addressed:
the important application of the information generated from this study for “be aguide to establish conservation priorities as well as efficient management programs. (Line 43-44)” was only poorly addressed in the discussion (Line 462-467) and need to be improved substantially. For example, what management strategy should be adapted for the expansion of ungulate populations in the Qinling Mountains? Should the expansion be encouraged by habitat management (e.g., the establishment of ungulate corridors)? Or their expansions should be suppressed to reduce the possible impacts to the local ecosystem and surrounding farmlands? And how to do it based on the evidence (e.g., their spatial distribution patterns) provided by this study, any suggestions?
Response: The coexistence of species is actually in dynamic coexistence, coexistence mechanism is very complex, and we can do at present is to just describe the coexistence state. It is beyond our current knowledge how to respond to patterns, many populations (e.g. giant panda, golden monkey, forest must deer, Mainland serow) have been low for many generations and are only now beginning to increase in our study area, we also don’t fully understand what management actions might be most effective to explosive growth of ungulate (e.g. wild boar, reeves muntjac, golden takin) in Qingling Mountains, the best course of action maybe “learn by doing”. Thus, long-term standardized monitoring programs on wildlife and their habitat will be necessary. Thus, we added a paragraph to the discussion section:
Carnivores played important roles in structuring communities (Steinmetz et al. 2020), as the removal of top predators may contribute to the population explosion of ungulates. The growth or expansion of ungulate species may have unexpected impact on previously established populations of coexistence, as well as a variety of effect on forest ecosystems, many of which were not fully understood. China had launched an ambitious program to establish a national park system, integrating current protected areas to resolve the problems of fragmented management (Xu et al. 2019). This park provided a unique opportunity to coordinate strategies and polices to restore large carnivore populations at local and landscape levels (Li et al. 2020). However, it wasn’t fully understand how ungulates would respond to current strategies and what management measures would be more effective to facilitate the recovery of large carnivore populations. Understanding the ecology of ungulate prey was important to predator conservation (Tan et al. 2018). Thus, long-term standardized monitoring programs on ungulates and their habitats will be necessary. Such programs could provide evidence-based insight into the direct impacts of future large predator restoration (although might require decades for the goal fulfilment). In addition, tolerance among local people and engaging policy makers to support the establishment of national parks along with targeted conservation monitoring was critical, so as to ensure that wildlife and people could co-occur in this area.
- the discussion on the mechanisms of coexistence (i.e., interspecific competition among ungulate species), as a theme of this article, was an over-interpretation of the data. Implication of the existing of interspecific competition requires at least evidence of the differences between the historical and present patterns in spatial and temperate distribution, or, better, differences of these distribution patterns between the studied population and the other populations with different community structure. There were no evidence supports this argument in this article, my suggestion is to remove all arguments concerning competition from this article.
Response: We have accepted this suggestion and made appropriate changes of title to weaken the competition theme of this article, present theme is to just describe temporal and spatial activity patterns of sympatric wild ungulates in Qinling Mountains, China
- Line 369-372: the attribution of the differences in annual activity patterns of sympatric ungulate species to minimize competition require solid evidence or
Response: OK, we added references here.
The results showed significant differences in peaks of annual activity patterns of wild ungulates, for such patterns may link the abundance and quality of food resources in different seasons, as well as temporal adjustment to minimize competition among sympatric wild ungulates [15,43,64].
[15] Li, J.; Xue, Y.D.; Zhang, Y.; Dong, W.; Shan, G.Y.; Sun, R.Q.; Hacker, C.; Wu, B.; Li, D.Q. Spatial and temporal activity patterns of Golden takin (Budorcas taxicolor bedfordi) recorded by camera trapping. PeerJ 2020, 8, e10353.
[43] Jia, X.D.; Liu, X.H.; Yang, X.Z.; Wu, P.F.; Songer, M.; Cai, Q.; He, X.B.; Zhu, Y. Seasonal activity patterns of ungulates in Qinling Mountains based on camera-trap data. Biodivers. Sci. 2014, 22, 737-745.
[64] Li, P.; Zhang, Z.J.; Yang, H.; Wei, W.; Zhou, H.; Hong, M.S.; Fu, M.X.; Song, X.Q.; Yu, J. Study on the activity rhythms of ungulates in Daxiangling Nature Reserve based on infrared camera trapping. J. Sichuan Forest. Sci. Technol. 2021, 42, 18-23.
- Line 61-70 in the Introduction is not relevant to this study.
Response: Although reviewer feel Line 61-70 should be removed, we request to keep these lines, as it serves as both an introduction to the role of ungulates in ecosystem, and a brief summary of potentially sources of conflicts between humans and wild ungulates. These lines serve as a link between the preceding and the following in the paper, these echoes the research content and background below. Thus, we recommend keeping these lines.
Carnivores played important roles in structuring communities [84], as the removal of top predators may contribute to the population explosion of ungulates. The growth or expansion of ungulate species may have unexpected impact on previously established populations of coexistence, as well as a variety of effect on forest ecosystems, many of which were not fully understood. China had launched an ambitious program to establish a national park system, integrating current protected areas to resolve the problems of fragmented management [85]. This park provided a unique opportunity to coordinate strategies and polices to restore large carnivore populations at local and landscape levels [86]. However, it wasn’t fully understand how ungulates would respond to current strategies and what management measures would be more effective to facilitate the recovery of large carnivore populations. Understanding the ecology of ungulate prey was important to predator conservation [87]. Thus, long-term standardized monitoring programs on ungulates and their habitats will be necessary. Such programs could provide evidence-based insight into the direct impacts of future large predator restoration (although might require decades for the goal fulfillment). In addition, tolerance among local people and engaging policy makers to support the establishment of national parks along with targeted conservation monitoring was critical, so as to ensure that wildlife and people could co-occur in this area.
- Line 444-449: hard to understand the meaning of this sentence
Response: we apologized for not being clear. Here we made appropriate changes:
The success of giant panda conservation in the study area appeared to have benefited sympatric wild ungulate species. For example, with the exception of the wild boar, previous studies have documented that wild ungulates were rarely found below 1,360 m in study area (ecotone of forestry and agriculture [44,50]). However, our camera traps detected most golden takins, mainland serow, reeves muntjac, tufted deer and Chinese goral at lower elevation zones (less than 1,360 m). This change may be due to the implementation of conservation programs, having returned low-elevation farmland to forests.

Reviewer 2 Report
Dear authors,
look at the few minor revisions I left in the document attached here.
Best, the reviewer

Author Response
Dear Reviewer:
We would like to thank Animals for giving us the opportunity to revise our manuscript again. Thank you for your comments on our manuscript. Those comments are very helpful for revising and improving our paper. We read all the comments carefully and tried our best to revise the manuscript, and have made much correction which we hope meet with approval. Below is our response to your comments.
Thanks for all the help
Best wishes,
Li Jia
#Reviewer
General comment (abstract): Letters of the words after (i), (ii), etc should not be capitalized. Same for “and” at line 41. Please correct
Response: Thank you. Change made.
Abstract: Dramatic increases in populations of wild ungulates have brought a new ecological issue in the Qinling mountains. Information on species’ niche differentiation will contribute to a greater understanding of the mechanisms of coexistence, so as to ultimately benefit the conservation and management of ecological communities. Here in this study, camera trapping was used to investigate spatial and temporal activity patterns of sympatric wild ungulates in the Qinling Mountains of China, where top predators were virtually absent. We obtained15,584 independent detections of seven wild ungulate species during 93,606 camera-trap days from April 2014 to October 2017. Results showed that (i) the capture rate differed significantly across species, with the capture rate of reeve muntjac significantly higher than that of other species; (ii) the wild boar had a higher occupancy rates (ψ=0.888) than other six ungulates, and distance to settlements had a negative relationship with wild boar (β=-0.24±0.17); (iii) the forest musk deer and mainland serow had low spatial overlaps with other five wild ungulates, while spatial overlap indices of any two given pairs of wild ungulates were relatively high; (iv) all wild ungulates species (expect wild boar) were mainly active during crepuscular and diurnal periods, and showed bimodal activity peaks at around 05:00-07:00 and 17:00-19:00; and finally, (v) all wild ungulates showed moderate to high temporal overlaps. The results provided detailed information of the spatial and temporal ecology of wild ungulate communities in forest ecosystems of China, which also would be a guide to establish conservation priorities as well as efficient management programs.
Line 138: What “approximate” means? Write as (April-October) and (November-March).
Response: Here we made change:
Based on the local climate characteristics, we divided the year into warm season (April-October) and cold season (November-March).
Line 144: Use present tense.
Response: OK, Change made.
Line 143: Please pay attention to the font of the sentence. The font must be always the same throughout the whole manuscript. Same line 191 and 302.
Response: Thank you, we carefully checked our manuscript.
Line 197: Remove the underlined font for the internet link. Thanks
Response: OK.
Line 244: Please be coherent with the other writing style when writing 0.5 Ë‚Δ ≤0.75
Response: Here we made change:
Activity overlap values were evaluated, with respect to the overall pairwise comparisons performed, considering Δ value ≤ 0.5 as low overlaps, 0.5Ë‚ Δ≤ 0.75 as moderate overlaps, and Δ > 0.75 as high overlaps.
Line 223: The citation at this line (i.e., Mella-Méndez et al. 2019) cannot be found in the bibliography. Please add it and be sure that ALL the citations within the ms are also listed in the reference section.
Response: Thank you. We added (Mella-Méndez et al. 2019) and checked all reference.
[55] Mella-Méndez, I.; Flores-Peredo, R.; Pérez-Torres, J.; Hernández-González, S.; González-Urbie, D.U.; Bolívar-Cimé, B.S. Activity patterns and temporal niche partitioning of dogs and medium-sized wild mammals in urban parks of Xalapa, Mexico. Urban ecosyst. 2019, 22, 1061-1070.

Reviewer 3 Report
Dear Authors,
I have read your paper with interest, but I have some remarks, necessary to change, especially References.
Remarks line by line:
9 - why dot after China? Why space between code and China?
46 - add a 'camera trapping' keyword
51 - way of cited references. In my opinion it is a lack of respect to MDPI and reviewer. Each paper should be submitted according to requirements. You should do the same like all other Authors. Of course, it is more comfortable to do it later or even not to do it in case of rejection by MDPI, but I don't understand why should we treat you better?
Even it that way it is such a mess in citing, one time coma, one not. For me lack of respect, anywhy.
MDPI requirements: e.g. [1,2]. etc....
59 - without the
66 - vegetation, without the
67 - without the
108 - too big space
149 - add a scale bar on a map. It is necessary. Change in a legend: Cells into cells
151 - put 'm' in 150 line.
197-198 - '30 m' should be in the same line
229-232 - something is different. Size? Breaks between lines?
296 - Table 3. Put a dot.
299-300 - between table and figure it should be a text. Think about thinner lines between lines in Table.
301 - a legend - everywhere you use "cells" - here blocks. Is there a difference? If not, use a cells consequently.
302 - Figure 2.
322 - you mean not May of one year, e.g. only 2014, but from period between 2014-2017, so 2014, 2015, 2016 and 2017? 4 x May month? It should be mentioned somewhere.
334- Figure 3 - add name of x axis. Tufted deer' chart is without numbers of x axis.
486-490 - these are not conclusions, but repeated statements from other part of the text. Conclusion is not a summary of repeated results. Please check definition of scientific conclusions. Please improve.
495 - why smaller letter? Moreover, is it really patents or Author Contributions? Another proof that you did not prepared you text as MDPI require. Use initial and surname, not full name and surname.
Page 18 - almost each subsection has got different breaks between lines. It should not look like that.
From 532 till the end of the text - you did not prepared references as MDPI require. What was the reason? Probably you have prepared this to another journal, in which the text was rejected? It decreased an overall merit. If your text will accept you have to change all references style:
Johnson, C.J.; Seip, D.R.; Boyce, M.S. A quantitative approach to conservation planning: using resource selection functions to 432
map the distribution of mountain caribou at multiple scales. Journal of Applied Ecology 2004, 41, 238-251.
Moreover, we put it in order of citing, not in alphabet! Lack of respect to reviewers' time and involvement. I'm dissapointed.
Author Response
Dear Reviewer:
We would like to thank Animals for giving us the opportunity to revise our manuscript again. Thank you for your comments on our manuscript. Those comments are very helpful for revising and improving our paper. We read all the comments carefully and tried our best to revise the manuscript, and have made much correction which we hope meet with approval. Below is our response to your comments.
Thanks for all the help
Best wishes,
Li Jia

This manuscript is a resubmission of an earlier submission. The following is a list of the peer review reports and author responses from that submission.
Round 1
Reviewer 1 Report
Revision animals-1695413:
Dear authors, the manuscript is interesting, methods appropriate and all the sections well-written. A few minor changes are required. Please correct following these recommendations. Best, the reviewer
Line 100: Substitute “is” with “are”.
Lines 102-105: What is this classification? Add classification by IUCN Red List please. Same lines 140-142. Cite IUCN Red List properly.
Line 109: Remove the first bracket and put a comma: “virtually absent, with the common leopard…”
Line 120: “which is virtually free…”. The “is” is missing.
Line 135: Names of seasons must be written with capital letter in English (e.g., Spring, Summer…). Please correct. Thanks
Lines 135-136: Be coherent, use Mar-May OR Jun to Aug style. Not a mixture.
Line 140: Which species is Rhinopithecus roxellarae? I suppose it is misspelled. Correct one Rhinopithecus roxellana?
Line 180: Missing citation. Provide ad least an example.
Line 199: Classify them as diurnal, nocturnal, crepuscular, cathemeral following what described by Gómez et al., 2005; Azevedo et al., 2018; Houngbégnon et al., 2020.
- Gómez, H., Wallace, R. B., Ayala, G., & Tejada, R. (2005). Dry season activity periods of some Amazonian mammals. Studies on Neotropical Fauna and Environment, 40(2), 91–95. https://doi.org/10.1080/01650520500129638
- Azevedo, F. C., Lemos, F. G., Freitas-Junior, M. C., Rocha, D. G., & Azevedo, F. C. C. (2018). Puma activity patterns and temporal overlap with prey in a human-modified landscape at Southeastern Brazil. Journal of Zoology, 305(4), 246–255. https://doi.org/https://doi.org/10.1111/jzo.12558
- Houngbégnon, F. G. A., Cornelis, D., Vermeulen, C., Sonké, B., Ntie, S., Fayolle, A., Fonteyn, D., Lhoest, S., Evrard, Q., Yapi, F., Sandrin, F., Vanegas, L., Ayaya, I., Hardy, C., Le Bel, S., & Doucet, J.-L. (2020). Daily Activity Patterns and Co-Occurrence of Duikers Revealed by an Intensive Camera Trap Survey across Central African Rainforests. Animals : An Open Access Journal from MDPI, 10(12). https://doi.org/10.3390/ani10122200
This depends, as described in those papers, on the % of records collected during daytime, night, or around duck and dawn. Please check if you have followed this rigorous classification unless change, modify and adapt. Cite properly the papers used.
Line 215: Missing citation. Add Meredith & Ridout, 2021. Meredith, M., & Ridout, M. (2021). Overview of the overlap package. https://cran.r-project.org/web/packages/overlap/vignettes/overlap.pdf
Lines 219-220: The citation is correct but those are not the intervals used by Monterroso et al. 2014 that you have to used too. Temporal overlaps: “low” with Δ ≤ 0.50; “moderate” with 0.5 < Δ ≤ 0.75; “high” with Δ > 0.75. If you need also “very low” (i.e., Δ ≤ 0.35) cite Sogliani et al. 2021 (https://doi.org/10.1007/s10344-021-01505-2), while if you need “very high” (Δ > 0.90) cite Andreoni et al. 2021 (https://doi.org/10.1080/03949370.2020.1777211). In those cases you have to change the other intervals accordingly. Change results, tables and so on accordingly to the new intervals. Thanks
Lines 223 and 226: Missing citation.
General comment: Change Budorcas bedfordi in Budorcas taxicolor bedfordi throughout the ms.
Line 383: expect wild boar OR except wild boar? I suppose the latter. Please check
Line 428: Cannot understand “…to expect large carnivores…”. Please check
Line 460: Check “probabiligy” à typo?. Replace “influence” with “influenced”.
Line 461: “we” is repeated twice. Correct
Line 463: over all without space à overall.
Reviewer 2 Report
This article reported the result of a 3.5-year camera trapping project on the survey of the 7 co-existing ungulates in Qinling Mountains, central China; it also intended to apply competition and niche segregation theories to the spatial distribution pattern and temperate activity pattern of these species. Despite of the intensive data collection effort done by this project, however, detail description of the environment of the camera trapping sites needs to be provided, and the presentation of the survey results requires reconsidered also before this article can be considered for publication. Specifically:
- For the 7 ungulate species recorded and discussed in the present study, there must be some variations in their habitat preference. Therefore, in order to justify the sampling effort was distributed representatively or evenly in various habitats, so the spatial distribution pattern comparison among species was reasonable, this paper needs to provide habitat types (e.g., ridge/valley/riparian/hillside, forest/grassland/farmland, elevation distribution, etc.) of all camera trapping sites in the method as the background information of the sampling effort in various habitats.
- For temporal activity pattern, using camera trapping data to compare daily activity pattern among different species were appropriate, but it was not appropriate for comparison of their annual (or seasonal) activity patterns. The RAI used in the present study were not only influenced by the activity level of the species but also affected by the abundance of the species, therefore the annual RAI fluctuation patterns showed in this paper can also be affected by the elevational migration behavior of certain species.
- The result of the present study provided a good understanding, if camera trapping site sampling correctly, of the present distribution pattern and present daily activity pattern of the 7 co-existing hoofed species, but the discussion on the competition or niche segregation theory was far-fetched and over-interpretated without long-term monitoring information in Qinling Mountains.